# Large Language Models are Better Reasoners with Self-Verification

**Yixuan Weng**[1][*] **Minjun Zhu**[1,2][*]**, Fei Xia**[1,2]**, Bin Li**[3]**,**
**Shizhu He**[1,2][†]**, Shengping Liu**[4]**, Bin Sun**[3]**, Kang Liu**[1,2,5]**, Jun Zhao**[1,2]

[1] The Laboratory of Cognition and Decision Intelligence for Complex Systems, IA, CAS
[2] School of Artificial Intelligence, University of Chinese Academy of Sciences
[3] College of Electrical and Information Engineering, Hunan University
[4]Unisound, Beijing, China          [5]Shanghai Artificial Intelligence Laboratory
wengsyx@gmail.com, {shizhu.he, kliu, jzhao}@nlpr.ia.ac.cn

## Abstract

Recently, with the chain of thought (CoT) prompting, large language models (LLMs), e.g., GPT-3, have shown strong reasoning ability in several natural language processing tasks such as arithmetic, commonsense, and logical reasoning. However, LLMs with CoT require multi-step prompting and multi-token prediction, which is highly sensitive to individual mistakes and vulnerable to error accumulation. The above issues make the LLMs need the ability to verify the answers. In fact, after inferring conclusions in some thinking decision tasks, people often check them by re-verifying steps to avoid some mistakes. In this paper, we propose and prove that LLMs also have similar self-verification abilities. We take the conclusion obtained by CoT as one of the conditions for solving the original problem. By performing a backward verification of the answers that LLM deduced for itself, we can obtain interpretable answer validation scores to select the candidate answer with the highest score. Experimental results demonstrate that the proposed method can improve the reasoning performance on various arithmetic, commonsense, and logical reasoning datasets. Our code is publicly available at: https://github.com/WENGSYX/Self-Verification.

## 1 Introduction

The ability of reasoning in the process of thinking and decision-making is an essential aspect of human intelligence. Recently, chain of thought (CoT) prompting (Wei et al., 2022) has been a good way to solve the arithmetic, commonsense, and logical reasoning tasks with large language models (LLMs), which help the LLMs simulating the human thinking process when solving complex natural language processing (NLP) tasks. CoT guides LLMs to generate a series of intermediate reasoning steps to address complex problems rather than just predict

a final answer. This approach has been shown the advance performances on several challenging NLP tasks, even when using only a few or no training samples (Madaan et al., 2022; Saparov and He, 2022; Fu et al., 2022; Gu et al., 2023).

Although CoT can enable LLMs to solve complex reasoning tasks, it is highly sensitive to individual mistakes and vulnerable to error accumulation (Shen et al., 2021). If a tiny mistake occurs, it can change the meaning deviations of the whole statement (Xiao et al., 2022), leading to incorrect answers (Cobbe et al., 2021). That is especially problematic in using CoT for addressing multi-step precise reasoning (such as mathematical calculation). Due to the lack of the error correction mechanism, it is difficult for the LLMs to obtain correct results from the possible errors in multiple steps reasoning. Detecting and mitigating errors is essential to improve reasoning capabilities.

Previous methods resolve the above issue by training another verifier to evaluate the correctness of the model's output (Shen et al., 2021; Li et al., 2022). However, there are some drawbacks in these work. On the one hand, training a verifier requires a lot of human annotations and additional fine-tuned models, which limits its widespread use in other tasks and domains. On the other hand, the verifier fine-tuned by a language model is not easily explainable, making it difficult to assess the model's reliability based on its output scores. Therefore, **the challenge of obtaining a better reasoner based on the LLMs is to get a verifier that can avoid manual annotation and additional training**, so that it can be better extended and migrated to other fields and tasks.

To address this challenge and overcome the limitations of training verifiers, we propose utilizing LLMs as reasoners with self-verification for selecting better prediction results. In numerous decision-making tasks, humans often perform self-verification of inferred conclusions to mitigate mis-

---

[*] : These authors contribute this work equally; [†] : Corresponding author.

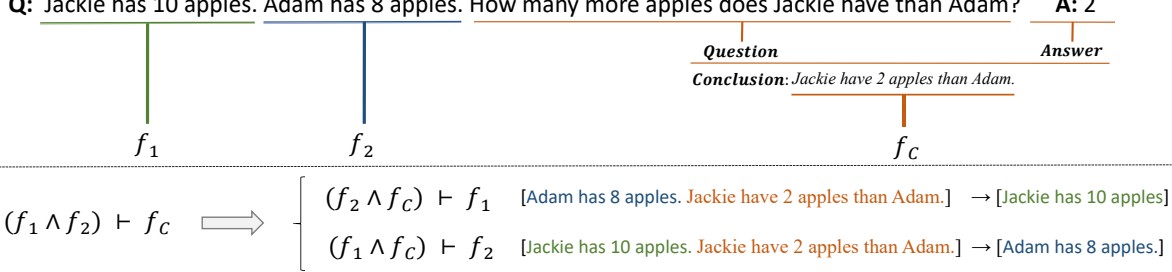

Figure 1: The answer of a question can be verified by masking and predicting the conditions of the original contexts. To mimic the self-verification ability of human, we predict the accuracy of $f_{\mathcal{C}}$ by predicting the original conditions $f_1$ or $f_2$ is right or not based on this conclusion.

takes (Poole and Mackworth, 2010). In this paper, we propose and demonstrate that LLMs possess a similar self-verification ability, the better reasoning with CoT is carried out in the following two steps, **Forward Reasoning** and **Backward Verification**. Specifically, in Forward Reasoning, LLM reasoners generate candidate answers using CoT, and the question and candidate answers form different conclusions to be verified. And in Backward Verification, We mask the original condition and predict its result using another CoT. We rank candidate conclusions based on a verification score, which is calculated by assessing the consistency between the predicted and original condition values. For example, as shown in Figure 1, by taking $f_2$ and $f_{\mathcal{C}}$ as conditions to predict the value of condition attribute in $\hat{f}_1$, the correctness of $f_{\mathcal{C}}$ can be evaluated by comparing the consistency of values of the predicted $\hat{f}_1$ and the original $f_1$ in verification.

Our method employs LLMs for self-verification with only a few prompts, eliminating the need for fine-tuning or gradient updating. This approach enables automatic verification of multiple candidate answers and corresponding conclusions, mitigating deviations from the correct thought chain in the original CoT. Our verification score arises from evaluating each step during the backward verification phase, rather than from the direct output of a neural network model (Cobbe et al., 2021; Li et al., 2022), enhancing the explainability of prediction outcomes and solution processes (Li et al., 2021; Yu et al., 2023; Zhu et al., 2023). We conducted experiments on various open-source datasets for mathematical reasoning, common sense, and logical reasoning tasks, achieving results beyond the baseline (e.g., $60.8 \rightarrow 65.1$ on GSM8K, $91.01 \rightarrow 93.40$ on SingleEq). In addition, we also attempt to combine our method with some approaches to improving

forward reasoning, such as self-consistency (Wang et al., 2023b) and Least-to-Most (Zhou et al., 2023). The experimental results show that our method also improves upon these forward reasoning approaches.

Our contributions are summarized as follows:

1. We propose and prove that large language models (LLMs) can self-verify their prediction results. The proposed method can provide interpretable verification scores without the need for train additional verifiers.

2. We have conducted extensive of experiments with multiple LLMs, and the experimental results on multiple mathematical, commonsense, and logical reasoning datasets achieve a significant improvement compared to the baseline.

3. We introduced True-False Item Verification for General Tasks in the backward verification stage and proposed Condition Mask Verification based on the characteristics of Arithmetic Tasks. Our method can be applied to a wide range of reasoning datasets, potentially paving the way for self-validation to become a new paradigm following pre-training and prompt learning, thus motivating further exploration of the capabilities of LLMs.

## 2 Related Work

**Language Model Reasoning**. It has been extensively studied in order to evaluate the various reasoning abilities of language models (Arora et al., 2022; Madaan et al., 2022; Sun et al., 2022), including arithmetic reasoning (Koncel-Kedziorski et al., 2015; Roy and Roth, 2016; Patel et al., 2021; Cobbe et al., 2021), commonsense reasoning (Talmor et al., 2018; Bhagavatula et al., 2019; Geva

et al., 2021; Zhu et al., 2022b), and logical reasoning (Liu et al., 2020; Yu et al., 2020). To solve these reasoning tasks, researchers have utilized pre-trained language reasoning models (Asai and Hajishirzi, 2020; Deng et al., 2021; Xia et al., 2022) or fine-tuned general LLMs (Cobbe et al., 2021). Early work attempted to solve complex reasoning tasks using Seq2Seq models (Wang et al., 2018; Li et al., 2019). Later, specialized encoder-decoder architectures were designed to improve reasoning performance (Shen and Jin, 2020; Zhu et al., 2022a). More recent work has suggested to adopt pre-training tasks to improve arithmetic reasoning ability (Yoran et al., 2021; Wang et al., 2022b). However, these methods require a significant amount of human annotation. In this paper, we proposed to obtain answers automatically and verify them in multiple reasoning tasks.

**In-context Learning**. Large language models such as GPT-3 exhibit impressive few-shot learning ability (Lu et al., 2022; Qiao et al., 2022), and closely approximate the predictors computed by gradient descent (Akyürek et al., 2022). It requires only filling a few exemplars into context as prompts and without the need for finetuning on a dataset of training examples (Wang et al., 2022a; Weng et al., 2023a). However, this approach struggles with tasks requiring complex reasoning (Rae et al., 2021), which drives researchers to explore other prompting strategies. CoT (Wei et al., 2022) is a chained reasoning approach that inserts a multi-step reasoning path before generating the final answer. Wang et al. (2023c) proposed a self-consistency decoding strategy to vote on the reasoning path, and Kojima et al. (2022) demonstrated that LLMs could as zero-shot reasoners through the prompt "Let's think step-by-step". These methods focus on constructing the CoT but ignore the high sensitivity of LLMs to individual mistakes in generating these chains, so some of these conclusions by CoT may be unreliable (Dhuliawala et al., 2023; Chu et al., 2023; Weng et al., 2023b). In this paper, we proved that LLMs can self-verify their conclusions.

**Answer Verification**. It is a common method for evaluating and reordering candidate answers with a trained language understanding model. Kushman et al. (2014) train a classifier to select the best answer from candidate answers, while Roy and Roth (2016) train a global scoring model to guide the search process for better answers. Shen et al.

(2021) proposed the joint training of answer generation and rank with language model. Cobbe et al. (2021) and Lightman et al. (2023) fine-tunes language model as a verifier, which calculates token-level and solution-level verification scores for a predicate result. However, the above method all need additional annotations. In our work, we do not require training examples and can provide an explainable verification score.

## 3 The Proposed Method

The proposed method can be used to verify prediction results. As shown in Figure 2, the process mainly consists of two steps. The first step, forward reasoning, is similar to the normal CoT, except that multiple candidate answers are generated through sampling decoding. In the second step, we calculate the verification scores for each candidate's answer by the self-verification method, and the answer with the highest score is selected as the final answer.

### 3.1 Forward Reasoning

In forward reasoning, the LLM reasoners generate candidate answers with the chain of thought prompting. We augment the input with several CoT prompts similar to the original query and then send it to the LLM. The LLM then performs sampling decoding to generate multiple candidates for verification.

As shown in Figure 2, for a reasoning task, the large language model $\mathcal{LLM}$ is given a question $\mathcal{X}$ which is accompanied by a chain of thought set C. In few-shot setting, the whole prompt also contains other question-CoT prompt-answer tuples. The input $\mathcal{X}$ can be further subdivided into $\mathcal{X} = \{f_1, f_2, \ldots, f_R, q\}$, where each $f_i$ is a condition (fact), and $q$ is a question, both represented as natural language clause or sub-sentences.

Specifically, in order to generate step-by-step solutions with CoT, we followed Wei et al. (2022) and designed CoT prompt set C for the reasoning dataset (e.g., the GSM8K dataset), which contains $n$ samples, each sample has the question $\dot{\mathcal{X}}$, chain of thout $\dot{t}$, and the answer $\dot{y}$. These samples are used as the input of test-time. Each example in C is concatenated as a prompt:

$$C = (\dot{\mathcal{X}}_0, \dot{\mathbf{t}}_0, \dot{\mathbf{y}}_0); (\dot{\mathcal{X}}_1, \dot{\mathbf{t}}_1, \dot{\mathbf{y}}_1); \ldots; (\dot{\mathcal{X}}_n, \dot{\mathbf{t}}_n, \dot{\mathbf{y}}_n)$$

Therefore, $\mathcal{LLM}$ is required to follow the prompt of C to generate the chain of thought $\mathbf{t_{CoT}}$ before generating the final answer $\mathbf{y}$:

## Step1: Forward Reasoning

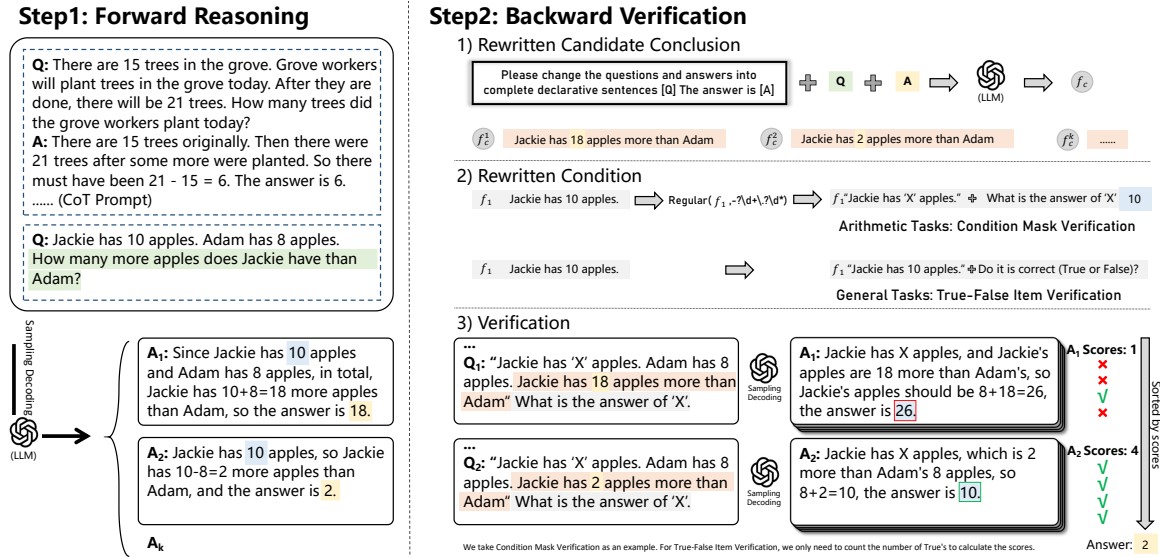

Figure 2: Example of self-verification. In the step one, LLM generates candidate answers and forms different conclusions. Then, in the step two, LLM verifies these conclusions in turn and computes the verification score.

$$\mathcal{P}(\mathbf{y}|C, \mathcal{X}) = \mathcal{P}(\mathbf{t_{CoT}}|C, \mathcal{X}) \times \mathcal{P})(\mathbf{y}|C, \mathcal{X}, \mathbf{t_{CoT}})$$

To ensure the diversity of different answers, we adapt sampling decoding (Radford et al., 2019) to generate multiple **y** for $K$ times. Specifically, sampling decoding is a random decoding method, which can select the next word by sampling from a probability distribution over the possible words at each step. Multiple candidate answers can be obtained when repeatedly using sampling decoding. For example, we generate *"18"* and *"2"* as candidate answers in the example of Figure 2.

### 3.2 Backward Verification

Step 1 may generate multiple different answers, this step is used to verify and select the best answer. Backward verification involves several sub-steps. First, the original question with each candidate's answer is rewritten as a conclusion and then supplemented as a new condition (incarnadine color in Figure 2). Then, we considered two methods to construct new questions. In the general QA task, the True-False Item Verification is given based on all the conditions, asking the LLM whether these conditions are mutually satisfied, it has a broad applicability. In Arithmetic reasoning tasks, as the definite condition masks can indicate the reasoning direction of the language model, we propose the Condition Mask Verification method to design questions for the verification stage. Finally,

we perform multiple experiments to compute the verification score by comparing the consistency between the predicted condition value and the original masked condition value, and select the candidate answer with the highest score as the final answer.

#### 3.2.1 Rewritten Candidate Conclusion

Besides, we rewrite the original question with the candidate's answer as a conclusion and then supplement it as a new condition in the backward verification step. Specifically, we use the instruction prompt *"Please change the questions and answers into complete declarative sentences [q] The answer is [y]"* to change $q$ and **y** into new declarative sentence $f_{\mathcal{Y}}$ by $\mathcal{LLM}$. As shown in Figure 2, we can rewrite the question and conclusion as *"Jackie has 18 apples more than Adam"*.

#### 3.2.2 Condition Masking

For question generation, the diversity of the problems makes it difficult to balance the need for coherence and fact consistency between questions and answers in practical operation (Sun et al., 2018; Ji et al., 2022). To tackle this issue, we included clear questions asking the language model to accurately predict.

**True-False Item Verification (TFV).** This approach can be applied to a wide range of reasoning QA tasks. We directly add "Do it is correct (True or False)?" after all the conditions, requiring the LLM to self-evaluate the correctness of these conditions.

**Condition Mask Verification (CMV).** Further,

we use regular expressions to filter out specific conditions, such as numbers, and then mask them in turn. If we do not mask all conditions but randomly select a condition, unnecessary conditions may be masked, which will significantly impact the verification answer. For example, "*Dana worked 9 hours on Friday, 10 hours on Saturday, and 3 hours on Sunday. She earns $13 per hour. How much money did Dana earn in weekend?*", since condition 1 (9 hours) does not affect the conclusion, it is difficult to predict it correctly. We replace all occurrences of $f$ in the original $\mathcal{X}$ with "X" in turn, and ask $\mathcal{LLM}$ to re-predict it. Then we rewrite the question. For example, we might find a value in $f_1$ and replace it with "X". We can then add "*What is the answer of 'X'?*" to the end of the new question, effectively turning it into an equation. This technique helps to guide the language model towards the correct answer.

### 3.2.3 Verification Score Calculation

This backward verification chain of thought is similar to solving an equation. We design a chain of thought prompt, like forward reasoning, to guide LLM in generating a solving process. We input the newly constructed sentences into $\mathcal{LLM}$. For TFV, we can directly count the number of answers that are True as the score, and for CMV, we will match its final result with the masked condition.

Due to the limited performance of LLM itself, if the condition is verified only once in the backward verification step, it is easy to have the same score, resulting in a lack of differentiation. To address this, we repeat the sampling decoding process $P$ times, so that the verification score can more accurately reflect the model's confidence for a given conclusion (Erd, 1970).

The verification score is calculated as follows:

$$\text{Score}_\mathbf{y} = \begin{cases} \sum_{p=1}^{P}(\sum_{r=1}^{R} 1_{(\mathcal{LLM}_p(\mathcal{X}-f_r+f_y)=f_r)}) & \text{TFV} \\ \sum_{p=1}^{P}(1_{(\mathcal{LLM}_p(\mathcal{X}+f_y))}) & \text{CMV} \end{cases}$$

Where $1_{(\bullet)}$ is an indicator function.

Finally, we select the one with the highest verification score from the K candidate answers generated as a result.

$$\mathbf{Output} = \underset{k \in [0,K]}{\text{argmax}}(\mathbf{Score}_k)$$

For example for CMV, in Figure 2.3)Verification, we match the results generated by the self-verification of LLM with the masked conditions.

There is one "*10*" in the conclusion of $A_1$, so the verification score is 1. There are four correct results in $A_2$, so the verification score is 4, and we finally choose $A_2$, which has the highest verification score, as the final conclusion.

## 4 Experiment Setting

### 4.1 Task and Dataset

We evaluated eight datasets on three reasoning tasks: arithmetic reasoning, commonsense reasoning, and logical reasoning. These datasets are highly heterogeneous in terms of their input formats (see Appendix A.2 for the detailed description of each dataset. Examples of different datasets are given in Table 7 of Appendix A.4).

- **Arithmetic Reasoning.** We performed experiments on the following 6 arithmetic datasets: SingleEq (Koncel-Kedziorski et al., 2015), AddSub (Hosseini et al., 2014), MultiArith (Roy and Roth, 2016), AQUA-RAT (Ling et al., 2017), GSM8K (Cobbe et al., 2021), and SVAMP (Arkil et al., 2021).

- **Commonsense Reasoning.** CommonsenseQA (CSQA) (Talmor et al., 2018) is the most typical dataset of the task, which requires commonsense knowledge about the world to accurately answer questions with complex meanings.

- **Logical Reasoning.** Date Understanding (DU) (Srivastava et al., 2022) involves inferring a date from a given context.

### 4.2 Model

We conducted experiments to evaluate the original GPT-3 (Chen et al., 2021) (code-davinci-001) model and the Instruct-GPT model (Ouyang et al., 2022) (code-davinci-002). Additionally, we conducted analysis experiments with public GPT-3 (Brown et al., 2020). All prediction results of different reasoning tasks and datasets are obtained by OpenAI's API [1]. Appendix A.3 shows the reproducibility statement.

### 4.3 Prompts

We conducted all experiments in the few-shot setting without any fine-tuning of the original LLM To ensure a fair comparison, we used the same

---

[1]OpenAI's API: https://openai.com/api/

| Method | | GSM8K | SingleEq | AddSub | MultiArith | AQUA-RAT | SVAMP | CSQA | DU |
|---|---|---|---|---|---|---|---|---|---|
| | | \multicolumn Arithmetic Tasks | | | | | | General Tasks | |
| Previous SOTA (Fine-tune) | | $35^a/57^b$ | $32.5^c$ | $94.9^d$ | $60.5^e$ | $37.9^f$ | $57.4^g$ | $91.2^h$ | - |
| 9–12 year olds | | $60^i$ | - | - | - | - | - | - | - |
| GPT-3 Standard | | 19.7 | 86.8 | 90.9 | 44.0 | 29.5 | 69.9 | 82.3 | 49.0 |
| GPT-3 (175B) `code-davinci-001` | CoT | 13.84 | 60.20 | 58.55 | 45.85 | 18.90 | 38.42 | 46.75 | 38.72 |
| | CoT+**Self-Verification** | **13.92**$_{(+0.08)}$ | **60.61**$_{(+0.41)}$ | **59.07**$_{(+0.52)}$ | **46.19**$_{(+0.34)}$ | **27.04**$_{(+8.14)}$ | **38.96**$_{(+0.54)}$ | **47.68**$_{(+0.93)}$ | **39.03**$_{(+0.31)}$ |
| Instruct-GPT (175B) `code-davinci-002` | CoT | 60.81 | 91.01 | 82.78 | 96.13 | 45.30 | 75.87 | 77.42 | 65.43 |
| | CoT+**Self-Verification** | **65.14**$_{(+4.33)}$ | **93.40**$_{(+2.39)}$ | **86.33**$_{(+3.55)}$ | **99.15**$_{(+3.02)}$ | **47.95**$_{(+2.65)}$ | **76.99**$_{(+1.12)}$ | **77.83**$_{(+0.41)}$ | **66.57**$_{(+1.14)}$ |
| \multicolumn Self-Consistency Decoding (Wang et al., 2023c) For Forward Reasoning | | | | | | | | | |
| GPT-3 (175B) `code-davinci-001` | SC | 23.40 | 70.25 | 68.65 | 79.82 | 25.60 | 54.58 | 54.92 | 49.26 |
| | SC+ **Self-Verification** | **23.59**$_{(+0.19)}$ | **70.50**$_{(+0.25)}$ | **68.71**$_{(+0.06)}$ | **80.01**$_{(+0.19)}$ | **28.98**$_{(+3.38)}$ | **54.68**$_{(+0.1)}$ | **55.09**$_{(+0.17)}$ | **49.72**$_{(+0.46)}$ |
| Instruct-GPT (175B) `code-davinci-002` | SC | 78.00 | 96.78 | 91.64 | 100.0 | 52.01 | 86.77 | 81.43 | 71.58 |
| | SC+**Self-Verification** | **78.32**$_{(+0.32)}$ | **96.85**$_{(+0.07)}$ | **92.03**$_{(+0.39)}$ | **100.0**$_{(+0.0)}$ | **52.25**$_{(+0.24)}$ | **86.94**$_{(+0.17)}$ | **81.53**$_{(+0.1)}$ | **71.89**$_{(+0.31)}$ |
| \multicolumn PAL (Gao et al., 2023) For Forward Reasoning | | | | | | | | | |
| GPT-3 (175B) `code-davinci-001` | PAL | 31.82 | 63.98 | 63.15 | 61.52 | 30.56 | 42.69 | - | - |
| | PAL+**Self-Verification** | **32.87**$_{(+1.05)}$ | **65.45**$_{(+1.47)}$ | **64.15**$_{(+1.0)}$ | **61.76**$_{(+0.24)}$ | **30.90**$_{(+0.34)}$ | **42.78**$_{(+0.09)}$ | - | - |
| Instruct-GPT (175B) `code-davinci-002` | PAL | 72.02 | 96.08 | 92.64 | 99.15 | 59.75 | 79.45 | - | - |
| | PAL+**Self-Verification** | **72.89**$_{(+0.87)}$ | **96.52**$_{(+0.44)}$ | **93.78**$_{(+1.14)}$ | **99.87**$_{(+0.72)}$ | **60.21**$_{(+0.46)}$ | **80.24**$_{(+0.79)}$ | - | - |

Table 1: Problem solve rate (%) on reasoning datasets. The previous SoTA results (baselines) are respectively obtained from: (a) GPT-3 175B finetuned (Cobbe et al., 2021); (b) GPT-3 175B finetuned plus an additional 175B verifier (Cobbe et al., 2021); (c) Hu et al. (2019); (d) Roy and Roth (2016); (e) Roy and Roth (2016); (f) Amini et al. (2019); (g) Pi et al. (2022); (h) Xu et al. (2022); (i) Cobbe et al., 2021). In addition, we also attempted to use self-consistency (SC) (Wang et al., 2023c) or PAL (Gao et al., 2023) (Since this method uses extra programs to replace the results of numerical calculations, we mainly compare it in Arithmetic Tasks.) Decoding to generate candidate answers during the Forward Reasoning stage and combine it with Self-Verification.

prompts as in Wei et al. (2022) for forward reasoning. We made several changes of the prompts for backward verification (the details are shown in Appendix A.5).

## 4.4 Implementation

In each experiment, we perform CoT prompting on the LLMs, then LLMs generate conclusions (answers) by sampling decoding without top-k truncation. When forward reasoning, we generated $K = 5$ candidate answers (conclusions). In backward verification, each candidate conclusion generated $P = 10$ times, and the maximum token length of each decoding was 168. After LLM generates the output, we only select the part of the text that conforms to the conclusion format. Appendix A.1 shows the specific strategy for different tasks. In addition, to ensure a fair comparison, we ran each experiment three times and calculated the average result.

## 5 Result and Analysis

The main experimental results are shown in Table 1. The table shows that the proposed self-verification method (SV) can improve previous methods in all datasets. Our method achieved a new state-of-the-art (SOTA) performance in six of these eight datasets. Appendix A.4 shows specific examples of language model self-verification for each dataset. Additionally, we observed that self-verification led to an average increase of 2.33% in the high-performing Instruct-GPT model, which indicates that the model with strong forward reasoning capabilities also benefits from the self-verification mechanism. The detailed experimental conclusions and analysis are described as follows:

**The current self-verification method is more suitable for arithmetic reasoning tasks than other reasoning tasks.** We find that the average performance improvement of arithmetic reasoning tasks (1.67%/2.84% ↑) is higher than that of other reasoning tasks (0.62%/0.78% ↑) in Table 1. We believe the reason is that it is easier to find the required mask conditions for arithmetic reasoning

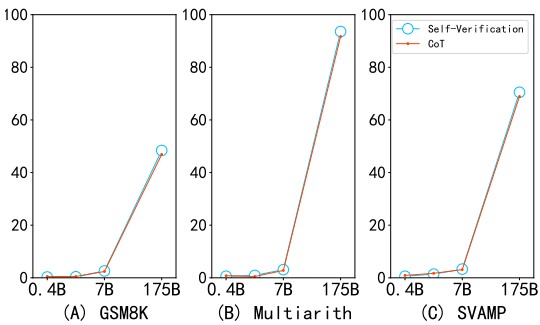

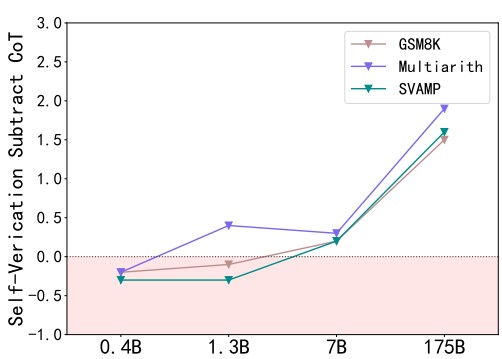

(a) Problem solve rate (%) in difference size models. The `text-ada-001` (0.4B), `text-babbage-001` (1.3B), `text-curie-001` (7B) and `text-davinci-002` (175B) models are used respectively.

(b) Subtract the problem solve rate (%) of CoT from the problem solve rate (%) of self-verification in different size models. The pink area means that the use of self-verification has a negative impact.

Figure 3: The self-verification ability of models with different sizes.

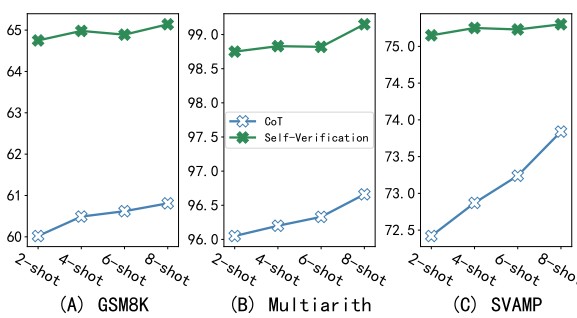

Figure 4: Problem solve rate (%) comparison of 2-shot to 8-shot prompts.

tasks, but other reasoning tasks used TFV that cannot determine the exact conditions. In the future, we will consider the targeted condition selection and masking for other reasoning tasks.

**The self-validation method can be combined with improved methods for forward reasoning.** We report the results of combining self-consistency or PoL at the bottom of Table 1 separately. Specifically, for combining self-consistency, we use the Top-2 candidate results obtained from self-consistency in the Forward Reasoning stage and then use self-validation to re-rank the candidate results; for combining PAL, we require the generation of runnable programs in Forward Reasoning to obtain candidate answers. We find that this approach still can achieve better performance than self-consistency, demonstrating that self-verification can be combined with a series of existing methods for improving forward calculation to achieve further gains. We believe that the self-

verification can re-rank candidate answers from the perspective of backward validation, providing more robust results.

**Larger language models are better reasoners with self-verification.** Figure 3 shows the capability of GPT-3 models with parameters ranging from 0.4B to 175B. The experimental results suggest that the self-verification ability tends to be more robust as the number of parameters increases, aligning with the limited experimental results in Wei et al. (2022). This indicates that self-verification is an emergent property that arises in larger models, where stronger reasoning capacities allow them to derive reliable and accurate conclusions during the context learning process (Ho et al., 2022; Wang et al., 2023a). Consequently, their performance in the backward verification phase is also enhanced. However, smaller models are prone to generate erroneous text during the generation process, and augment them with self-verification abilities presents a challenge.

**With the different number of few-shots, the reasoning ability of models using self-verification has significantly improved.** Figure 4 demonstrate the impact of different sample sizes on three arithmetic reasoning datasets. We observe that the self-verification method exhibits greater robustness with smaller samples, even with only 2-shots (At this time, it has 99.6% performance of 8-shot, while CoT has only 98.7%). In addition, we find that even with only four samples (2 CoT samples + 2 self-verification samples), self-verification outperforms the CoT with eight samples, which

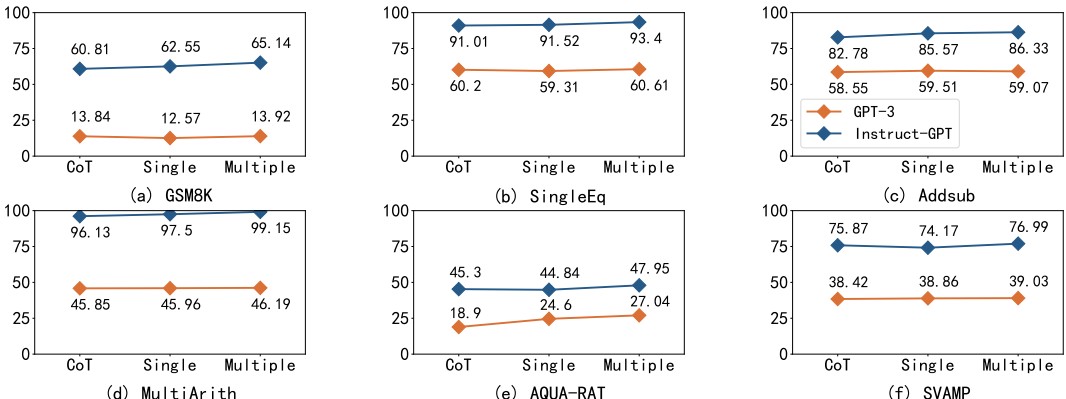

Figure 5: Comparison of problem solve rate (%) between single-condition verification and multiple-condition verification.

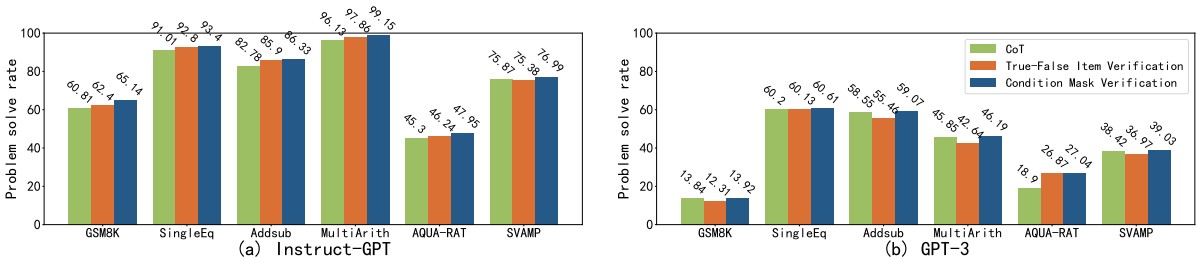

Figure 6: Comparison of problem solve rate (%) for the "CMV" and the "TFV" in arithmetic tasks.

highlights the importance of answer verification in scenarios of limited data.

**The more verification conditions are used, the better self-verification reasoning ability.** We observed the effect of using the single conditional mask on six different arithmetic datasets for Condition Mask Verificat in Figure 5. As each number in these datasets' input can be thought of as a condition, we can study the impact of increasing the number of validation conditions. In most experiments, we found that the multi-condition mask performed better than the single-condition mask, and both performed better than the original CoT. These results suggest that the accuracy of verification scores improves as the number of available conditions increases.

**Masked conditions can guide the LLMs to reason more effectively.** As shown in Figure 6, we compared the results of using CMV (Conditional Masked Verification) and TFV (Token Form Verification) for self-verification. We found that the performance of CMV is generally better than TFV. We believe this is because the lack of explicit goals can lead to a lack of use of existing conclusions, so CMV is more helpful in stimulating the self-

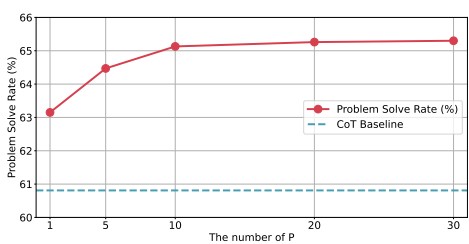

Figure 7: The computational resource of the proposed method on GSM8K.

verification ability of the model. However, due to its simplicity, TFV can be applied to a variety of tasks (including common sense reasoning and logical reasoning, both with improvements compared to the CoT baseline) for self-verification, making it highly adaptable to different scenarios.

**Fewer computational resources can also improve performance through self-verification.** In Figure 7, we show the results of changing the number of $P$ generated in Backward Verification. We find that even when $P = 2$, only a small increase in computational overhead is needed, and there is still an improvement in CoT baseline. Considering that performance starts to slowly increase when P is

increased to 10, we recommend choosing an appropriate value for P (e.g. P=10) to achieve a balance between performance and resource consumption.

| Dataset | Num. | Avg. Ans. | Ran. Acc. | CoT Acc. | Veri. Acc. |
|---------|------|-----------|-----------|----------|------------|
| GSM8K | 601/1306 | 2.80 | 35.7% | 53.4% | **58.9%** |
| Addsub | 77/377 | 2.13 | 46.9% | 56.7% | **74.0%** |
| AQUA | 123/219 | 2.71 | 36.9% | 50.3% | **51.2%** |
| Multiarith | 73/593 | 2.19 | 45.6% | 71.2% | **89.0%** |
| SingleEq | 51/501 | 2.31 | 43.3% | 51.0% | **74.5%** |
| SVAMP | 274/954 | 2.54 | 39.4% | 51.5% | **58.4%** |

Table 2: Further analysis of the experimental results in Table 1. Where "Num." represents the total number of samples in this setting and the total number of original dataset samples, "Avg. Ans." means the average number of candidate answers, "Ran. Acc." = 1 / "Avg. Ans.". "CoT Acc." refers to the accuracy of the CoT method when temperature = 0. And "Veri. Acc." refers to the probability of picking out the correct answer from the candidate answers in backward verification stage.

**Analysis of the accuracy of the verification results.** We conducted additional analysis of the InstructGPT results in Table 1 under a new setting where we only considered cases in the forward inference phase that contained one correct answer and $N$ incorrect answers (where $N>=1$ and $N<=4$). The results in Table 2 provide further evidence that the proposed self-verification technique can effectively improve the accuracy of commonsense reasoning models. Across all 6 datasets, the verification accuracy is consistently and considerably higher than both the random guessing baseline and the standalone CoT model accuracy. For example, on the challenging GSM8K dataset, the verification stage obtains 58.9% accuracy, substantially outperforming the 53.4% CoT accuracy and 35.7% random guess accuracy. The largest accuracy gains are witnessed on the MultiArith and SingleEq datasets, where the verification stage lifts the accuracy by 17.8% and 23.5% respectively over the CoT model. This indicates that the self-verification technique is particularly adept at rectifying errors made by the CoT model on arithmetic and symbolic equation problems. The consistent accuracy improvements demonstrate that allowing the model to verify its own predictions provides a simple yet effective way to enhance commonsense reasoning. These comprehensive results validate self-verification as a promising approach to refine LLMs and reduce reasoning errors.

## 6 Conclusion

In this study, we show that large language models have a strong ability to self-verification, allowing them to assess the conclusions they generate accurately. We propose a novel method that uses self-verification to generate interpretable scores for ranking results in few-shot tasks. Our approach demonstrates the potential of using self-verification to improve the accuracy and reliability of large language models in reasoning tasks. By relying on the self-verification ability of large language models, we significantly improved the accuracy of three types of reasoning tasks. All in all, we posit that the self-verification capability of large language models will have an extensive positive impact, as it enables the enhancement of their reasoning abilities through a simple process of self-verification.

## Limitations

Our self-verification method relies on large language models (LLMs). It provides few-shot prompts to guide the model in verifying its own results, but it is worth noting that these prompts are artificially constructed and may introduce bias. The effectiveness of our method is limited by the presence of accurate answers within the candidate conclusions generated by the LLM, our experiments demonstrate that the capability of self-verification depends on the model's reasoning abilities, which means it is challenging to augment the reasoning performance of smaller language models, while the benefits are greater for high-performing models (in Figure 3). For the sake of usability, our method focuses on the conclusions derived from the reasoning rather than the reasoning process itself, and is thus not suitable for evaluating the LLM's inference procedure. Additionally, the method necessitates generating multiple candidate inference chains and conclusions, leading to increased computational costs; however, we demonstrate that only a minimal increase (merely 1x) in expenditure is required to substantially enhance the model's inference capabilities (in Figure 7).

## Acknowledgements

This work was supported by the National Key R&D Program of China (No.2022ZD0118501) and the National Natural Science Foundation of China (No.62376270, No.U1936207, No.61976211, No.62171183). This work was supported by the Strategic Priority Research Program of Chinese Academy of Sciences (No.XDA27020100), Youth Innovation Promotion Association CAS, and OPPO Research Fund.

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

# A Appendix

## A.1 Answer Cleansing

Our answer cleaning strategy is consistent with Wang et al. (2023b) and Zhang et al. (2022). The first number after selecting *"The answer is"* is regarded as an output for arithmetic tasks, and we use Numpy (Harris et al., 2020) to compare it with the standard answer. For multiple choice tasks, we regard the first capital letter as output.

For the "True-False Item Verification", we use "True or False" to select answer. Table 3 summarizes a list of answer cleansing approaches used across all the experiments.

## A.2 Dataset Details

Our method is evaluated on eight benchmark datasets that cover arithmetic reasoning, commonsense reasoning, and logical reasoning tasks. The statistics of the datasets are shown in Table 6.

We list the details for all datasets used in this paper.

- **GSM8K**: https://github.com/openai/grade-school-math

- **SingleEq**: https://gitlab.cs.washington.edu/ALGES/TACL2015

- **AddSub**: https://www.cs.washington.edu/nlp/arithmetic

- **MultiArith**: http://cogcomp.cs.illinois.edu/page/resource_view/98

- **AQUA-RAT**: https://github.com/deepmind/AQuA

- **SVAMP**: https://github.com/arkilpatel/SVAMP

- **CSQA**: https://www.tau-nlp.org/commonsenseqa

- **Data Understanding**: https://github.com/google/BIG-bench

## A.3 Reproducibility Statement

All our experiments in the main text were run using the OpenAI API on November 20th to December 20th, 2022. The main experiment was run on November 25th to December 10th, the single-condition rxperiment was run on November 20th to 25th, the Few CoT prompts experiment was run on December 12th, the True-False Item Verification experiment was run on December 12th to 15th, the different sizes models experiment was run on December 16th, and the computational reasource experiment was run on December 18th.

## A.4 Additional Experiment Results

In Table 5, we show whether to generate real examples of multiple condition masks. We found that if only the first condition mask is used, the score is zero, and multiple evidence masks can obtain more accurate verification scores.

Then, Table shows the difference between (A) Conditional Masked Verification and (B) True-False Item Verification. The difference between the two lies in whether the condition mask is correct and the template for the question.

Finally, we generated some self-verification examples using the Instruct-GPT (code-davinci-002) model. As show in Table 7.

| Answer Format | Answer Cleansing Approach | Pseudo Code (Example in Pytorch 3.7) |
|---|---|---|
| Number | Pick up the first number encountered in the text. | ```pred = pred.replace(",", "")```
```pred = [s for s in re.findall(r'-?\d+\.?\d*', pred)]```
```pred = pred[0]``` |
| Multiple-Choice | Pick up the first large letter encountered in the text. | ```pred = re.findall(r'A|B|C|D|E', pred)```
```pred = pred[0]``` |
| True or False | Pick up the first "True" or "False" encountered in the text after removing unnecessary letters. | ```pred = pred.lower()```
```pred = re.sub("\"|\'|\n|\.|\s|\:|\,","_", pred)```
```pred = pred.split("_")```
```pred = [i for i in pred if i in ("True", "False")]```
```pred = pred[0]``` |
| Yes or No | Pick up the first "yes" or "no" encountered in the text after removing unnecessary letters. | ```pred = pred.lower()```
```pred = re.sub("\"|\'|\n|\.|\s|\:|\,","_", pred)```
```pred = pred.split("_")```
```pred = [i for i in pred if i in ("yes", "no")]```
```pred = pred[0]``` |
| Free Format | Just remove unnecessary letters, such as ".". | ```pred = re.sub("\"|\'|\n|\.|\s","", pred)``` |

Table 3: Detail description of answer cleansing.

Figure 8: Sample comparison of different verification methods

## A.7 True-False Item Verification prompt

In order to test True-False Item Verification, we designed some prompts to guide the LLMs to consider whether the conclusion is wrong.

As show in Table 16, Table 17.

## A.5 Forward reasoning prompt

For fairness, we used the same CoT prompts cue as Wei et al. (2022) in forward reasoning.

As show in Table 8, Table 9, Table 10, Table 11.

## A.6 Backward Verification prompt

When backward verification, we manually wrote some examples referring to Wei et al. (2022).

As show in Table 12, Table 13, Table 14, Table 15.

**EXAMPLE**

**Q:** Tim wanted to make lemonade for a pool party. For a gallon of lemonade, his recipe called for 1 cup of fresh lemon juice. He found that 6 lemons would yield 1 cup of juice. He figured he would need to make 4 gallons of lemonade for the party. His best friend Allen asked if Tim could make an extra gallon for him that was twice as tart as the other gallons. How many lemons will Tim need? **A: The answer is 36.** (That's the right answer.)

| True-False Item Verification | If Tim will need 36 lemons, and he will need to make 4 gallons of lemonade for the party, so he will need 4 * 6 = 24 lemons, he will need to make an extra gallonfor him that was twice as tart as the other gallons, so he will need 24 * 2 = 48 lemons,48 != 36, which is different from the theme. The answer is False. [✗] |
|---|---|
| Conditional Masked Verification | Tim wanted to make lemonade for a pool party. For a gallon of lemonade, his recipe called for X cup of fresh lemon juice. He found that 6 lemons would yield 1 cup of juice. He figured he would need to make 4 gallons of lemonade for the party. His best friend Allen asked if Tim could make an extra gallon for him that was twice as tart as the other gallons. Tim will need 36.0 lemons. X * 6 * 4 + X * 6 * 2 = 36, 36 * X = 36, X = 1. The answer is 1. [✓] |

Table 4: We have selected an actual generated examples to further demonstrate the impact of different verification methods.

**EXAMPLE 2** (Condition is not required)

**Q:** Dana worked 9 hours on Friday, 10 hours on Saturday, and 3 hours on Sunday. She earns $13 per hour. How much money did Dana earn in weekend?

**A:** Dana earns 13 dollars per hour. She worked 10 hours on Saturday, and 3 hours on Sunday. So she earned 13 * (10 + 3) = 13 * 13 = 169 dollars. The answer is 169. [✓]

| Condition 1 mask | Condition 2 mask | ... |
|---|---|---|
| **Q:** Dana worked 'X' hours on Friday, 10 hours on Saturday, and 3 hours on Sunday. She earns $13 per hour. Dana earn 169 money in weekend. | **Q:** Dana worked 9 hours on Friday, 'X' hours on Saturday, and 3 hours on Sunday. She earns $13 per hour. Dana earn 169 money in weekend. | ... |
| Separate score: 0 | Separate score: 8 | ... |
| Total verification score: 19 | | |

Table 5: This is an example of the need to use multiple Conditions.

| Dataset | Number of samples | Average words | Answer Format | Lience |
|---|---|---|---|---|
| GSM8K | 1319 | 46.9 | Number | MIT License |
| SingleEq | 508 | 27.4 | Number | MIT License |
| AddSub | 395 | 31.5 | Number | Unspecified |
| MultiArith | 600 | 31.8 | Number | Unspecified |
| AQUA-RAT | 254 | 51.9 | Multiple Choice | Apache-2.0 |
| SVAMP | 1000 | 31.8 | Number | MIT License |
| CSQA | 1221 | 27.8 | Multiple Choice | Unspecified |
| Data Understanding | 369 | 21.4 | Date | Unspecified |

Table 6: Dataset Description.

Table 7: Example outputs by self-verification for each datasets.

| Dataset | Question | Self-Verification |
|---|---|---|
| GSM8K | Q: Matthew has a collection of 12 unique toy soldiers. He wants to sell them for a fair price. He found a buyer who is willing to pay for half his collection $5 per toy, and for the other half 'X' per toy. If Matthew would agree to that offer, He would earn 72.0 dollars. | A: Matthew has 12 unique toy soldiers. He wants to sell them for a fair price. He found a buyer who is willing to pay for half his collection $5 per toy. and for the other half $X per toy. If Matthew would agree to that offer. He would earn 72.0 dollars. 12 / 2 = 6 6 * 5 = 30 12 - 6 = 6 6 * X = 42 X = 42 / 6 X = 7 The answer is 7. [✓] |
| | | Continued on next page |

| Dataset | Question | Self-Verification |
|---|---|---|
| | Q: Four students scored a total of 251 points in a board game. Naomi scored 68 of the points. Yuri scored 'X' more than half as many points as Naomi and Brianna scored 17 points more than Naomi. Jojo scored 54 points. | Four students scored a total of 251 points. Naomi scored 68 points. Yuri scored X points. Brianna scored 17 points more than Naomi. Jojo scored 54 points. So: 68 + X + 68 + 17 + 54 = 251, X = 251 - 68 - 68 - 17 - 54, X = 94. The answer is 94. **[✗](GT : 10)** |
| SingleEq | Q: Sam grew 4 watermelons, but the rabbits ate 'X' watermelons. Sam have 1 watermelon left. | A:Sam grew 4 watermelons. But the rabbits ate X watermelons. So, we can write the following equation: 4 - X = 1, X = 4 - 1, X = 3. The answer is 3. **[✓]** |
| | Q: Melanie, Benny, Sandy, and Jessica each have 'X' dozen Pokemon cards. They have 432 pokemon cards in all. | A: Melanie has X dozen Pokemon cards. Benny has X dozen Pokemon cards. Sandy has X dozen Pokemon cards. Jessica has X dozen Pokemon cards. So, they have 4 X dozen Pokemon cards in total. X * 4 = 432, X = 432 / 4, X = 108. The answer is 108. **[✗](GT : 9)** |
| Addsub | Q: Karin 's science class weighed plastic rings for an experiment . They found that the orange ring weighed 'X' ounce , the purple ring weighed 0.3333333333333333 ounce , and the white ring weighed 0.4166666666666667 ounce . The total weight of the plastic rings is 0.8333333333333334. | A: The orange ring weighed 0.08333333333333333 ounce. the purple ring weighed 0.3333333333333333 ounce. and the white ring weighed 0.4166666666666667 ounce. The total weight of the plastic rings is 0.8333333333333334. 0.08333333333333333 + 0.3333333333333333 + 0.4166666666666667 = 0.8333333333333334 0.8333333333333334 - 0.4166666666666667 = 0.4166666666666667 0.4166666666666667 - 0.3333333333333333 = 0.08333333333333334 The answer is 0.08333333333333334. **[✓]** |
| | Q: There were 6 roses in the vase . Mary cut some roses from her flower garden . There are now 'X' roses in the vase . She cut 10 roses. | A: There were originally 6 roses in the vase. Mary cut 10 roses from her flower garden. So there are now X roses in the vase.  6 - 10 = X X = -4  The answer is -4. **[✗](GT : 16)** |
| MultiArith | Q: Luke was organizing his baseball cards in a binder with 3 on each page. If he had 3 new cards and 'X' old cards to put in the binder, he use 4 pages. | A: Luke had 3 new cards and X old cards. He would use 4.0 pages. So, we can write the following equation: 3 + X = 3 * 4.0, 3 + X = 12, X = 12 - 3 X = 9. The answer is 9. **[✓]** |
| | Q: Will bought 7 boxes of chocolate candy and gave 3 to his little brother. If each box has 'X' pieces inside it, Will still have 16 pieces. | A: Will bought 7 boxes of chocolate candy. Each box has X pieces inside it. He gave 3 boxes to his little brother. So he has 4 boxes left. 7 * X = Y, Y - 3 * X = 16, Y = 16 + 3 * X, Y = 16 + 3 * X. The answer is 16. **[✗](GT : 4)** |

| Dataset | Question | Self-Verification |
|---|---|---|
| SVAMP | Q: Jake has 3 fewer peaches and 10 more apples than Steven. Steven has 'X' peaches and 15 apples. Steven has 3.0 more apples than peaches. | A: Jake has 3 fewer peaches and 10 more apples than Steven. Steven has X peaches and 15 apples. Steven has 3.0 more apples than peaches. X + 3 = 15, X = 15 - 3, X = 1.2 The answer is 12. [✓] |
| | Q: An industrial machine can make 6 shirts a minute. It worked for 5 minutes yesterday and for 'X' minutes today. The machine made 72 shirts today. | A: The machine can make 6 shirts a minute. It worked for 5 minutes yesterday and for X minutes today. The machine made 72 shirts today. So, we can write the following equation: 6 * 5 + 6 * X = 72, 30 + 6X = 72, 6X = 72 - 30, 6X = 42, X = 42 / 6 X = 7. The answer is 7. [✗](GT : 12) |
| Common-senseQA | Q: If there is a pond with trees around it, it likely located in forest. | A: he forest is likely located where there is a pond with trees around it. The answer is Yes. [✓] |
| | Q: The flasks was used to distill elements, it is used in chemistry lab. | A: The answer should be the place where the flask is used. But the chemistry lab is not suitable place where the flask is used. The answer is No. [✗](GT : Yes) |
| Date Under-stand. | Q: It was 'X'/01/2021 a week ago. Today is 09/08/2021. | A: If today is 09/08/2021, so the date a week ago is 09/01/2021. The answer is 09. [✓] |
| | Q: Yesterday, Jan 21, 2011, Jane ate 2 pizzas and 5 wings. One week ago from today is 01/15/2011. | A: one week ago from today is 01/15/2011. So today is 01/22/2011. So yesterday is 01/21/2011. So Jan 'X'1 is 01/21/2011. So 'X'011 is 2011. So The answer is 2011. [✗](GT : B) |

Table 8: Few-shot exemplars for full chain of thought prompt for AQuA.

---

**PROMPT FOR AQUA ALGEBRAIC WORD PROBLEMS**

**Q:** John found that the average of 15 numbers is 40. If 10 is added to each number then the mean of the numbers is?
Answer Choices: (a) 50 (b) 45 (c) 65 (d) 78 (e) 64

**A:** If 10 is added to each number, then the mean of the numbers also increases by 10. So the new mean would be 50. The answer is (a).

**Q:** If a / b = 3/4 and 8a + 5b = 22,then find the value of a.
Answer Choices: (a) 1/2 (b) 3/2 (c) 5/2 (d) 4/2 (e) 7/2

**A:** If a / b = 3/4, then b = 4a / 3. So 8a + 5(4a / 3) = 22. This simplifies to 8a + 20a / 3 = 22, which means 44a / 3 = 22. So a is equal to 3/2. The answer is (b).

**Q:** A person is traveling at 20 km/hr and reached his destiny in 2.5 hr then find the distance?
Answer Choices: (a) 53 km (b) 55 km (c) 52 km (d) 60 km (e) 50 km

**A:** The distance that the person traveled would have been 20 km/hr * 2.5 hrs = 50 km. The answer is (e).

**Q:** How many keystrokes are needed to type the numbers from 1 to 500?
Answer Choices: (a) 1156 (b) 1392 (c) 1480 (d) 1562 (e) 1788

**A:** There are 9 one-digit numbers from 1 to 9. There are 90 two-digit numbers from 10 to 99. There are 401 three-digit numbers from 100 to 500. 9 + 90(2) + 401(3) = 1392. The answer is (b).

Table 9: Few-shot exemplars for full chain of thought prompt for Date Understanding.

---

**PROMPT FOR DATE UNDERSTANDING**

**Q:** 2015 is coming in 36 hours. What is the date one week from today in MM/DD/YYYY?

**A:** If 2015 is coming in 36 hours, then it is coming in 2 days. 2 days before 01/01/2015 is 12/30/2014, so today is 12/30/2014. So one week from today will be 01/05/2015. So the answer is 01/05/2015.

**Q:** The first day of 2019 is a Tuesday, and today is the first Monday of 2019. What is the date today in MM/DD/YYYY?

**A:** If the first day of 2019 was Tuesday, then 01/01/2019 was a Tuesday. Today is the first monday, would be six days later. So today is 01/07/2019. So the answer is 01/07/2019.

**Q:** The concert was scheduled to be on 06/01/1943, but was delayed by one day to today. What is the date 10 days ago in MM/DD/YYYY?

**A:** One day after 06/01/1943 is 06/02/1943, so today is 06/02/1943. 10 days before today is 05/23/1943. So the answer is 05/23/1943.

**Q:** It is 4/19/1969 today. What is the date 24 hours later in MM/DD/YYYY?

**A:** Today is 04/19/1969. 24 hours later is one day after today, which would be 04/20/1969. So the answer is 04/20/1969.

**Q:** Jane thought today is 3/11/2002, but today is in fact Mar 12, which is 1 day later. What is the date 24 hours later in MM/DD/YYYY?

**A:** Today is 03/12/2002. So the date 24 hours later will be 03/13/2002. So the answer is 03/13/2002.

**Q:** Jane was born on the last day of Feburary in 2001. Today is her 16-year-old birthday. What is the date yesterday in MM/DD/YYYY?

**A:** The last day of February is the 28th, so Jane was born on 02/28/2001. Today is her 16-year old birthday, so today is 02/28/2017. So yesterday was 02/27/2017. So the answer is 02/27/2017.

Table 10: Few-shot exemplars for full chain of thought prompt for math word problems. This set of exemplars was used for all math word problem datasets except AQuA.

---

**PROMPT FOR MATH WORD PROBLEMS**

**Q:** There are 15 trees in the grove. Grove workers will plant trees in the grove today. After they are done, there will be 21 trees. How many trees did the grove workers plant today?

**A:** There are 15 trees originally. Then there were 21 trees after some more were planted. So there must have been 21 - 15 = 6. The answer is 6.

**Q:** If there are 3 cars in the parking lot and 2 more cars arrive, how many cars are in the parking lot?

**A:** There are originally 3 cars. 2 more cars arrive. 3 + 2 = 5. The answer is 5.

**Q:** Leah had 32 chocolates and her sister had 42. If they ate 35, how many pieces do they have left in total?

**A:** Originally, Leah had 32 chocolates. Her sister had 42. So in total they had 32 + 42 = 74. After eating 35, they had 74 - 35 = 39. The answer is 39.

**Q:** Jason had 20 lollipops. He gave Denny some lollipops. Now Jason has 12 lollipops. How many lollipops did Jason give to Denny?

**A:** Jason started with 20 lollipops. Then he had 12 after giving some to Denny. So he gave Denny 20 - 12 = 8. The answer is 8.

**Q:** Shawn has five toys. For Christmas, he got two toys each from his mom and dad. How many toys does he have now?

**A:** Shawn started with 5 toys. If he got 2 toys each from his mom and dad, then that is 4 more toys. 5 + 4 = 9. The answer is 9.

**Q:** There were nine computers in the server room. Five more computers were installed each day, from monday to thursday. How many computers are now in the server room?

**A:** There were originally 9 computers. For each of 4 days, 5 more computers were added. So 5 * 4 = 20 computers were added. 9 + 20 is 29. The answer is 29.

**Q:** Michael had 58 golf balls. On tuesday, he lost 23 golf balls. On wednesday, he lost 2 more. How many golf balls did he have at the end of wednesday?

**A:** Michael started with 58 golf balls. After losing 23 on tuesday, he had 58 - 23 = 35. After losing 2 more, he had 35 - 2 = 33 golf balls. The answer is 33.

**Q:** Olivia has $23. She bought five bagels for $3 each. How much money does she have left?

**A:** Olivia had 23 dollars. 5 bagels for 3 dollars each will be 5 x 3 = 15 dollars. So she has 23 - 15 dollars left. 23 - 15 is 8. The answer is 8.

---

Table 11: Few-shot exemplars for full chain of thought prompt for CSQA. There are newlines between the answer choices that are omitted in the table for space reasons.

---

**PROMPT FOR CSQA**

**Q:** What do people use to absorb extra ink from a fountain pen? Answer Choices: (a) shirt pocket (b) calligrapher's hand (c) inkwell (d) desk drawer (e) blotter

**A:** The answer must be an item that can absorb ink. Of the above choices, only blotters are used to absorb ink. So the answer is (e).

**Q:** What home entertainment equipment requires cable?
Answer Choices: (a) radio shack (b) substation (c) television (d) cabinet

**A:** The answer must require cable. Of the above choices, only television requires cable. So the answer is (c).

**Q:** The fox walked from the city into the forest, what was it looking for? Answer Choices: (a) pretty flowers (b) hen house (c) natural habitat (d) storybook

**A:** The answer must be something in the forest. Of the above choices, only natural habitat is in the forest. So the answer is (b).

**Q:** Sammy wanted to go to where the people were. Where might he go? Answer Choices: (a) populated areas (b) race track (c) desert (d) apartment (e) roadblock

**A:** The answer must be a place with a lot of people. Of the above choices, only populated areas have a lot of people. So the answer is (a).

**Q:** Where do you put your grapes just before checking out? Answer Choices: (a) mouth (b) grocery cart (c)super market (d) fruit basket (e) fruit market

**A:** The answer should be the place where grocery items are placed before checking out. Of the above choices, grocery cart makes the most sense for holding grocery items. So the answer is (b).

**Q:** Google Maps and other highway and street GPS services have replaced what? Answer Choices: (a) united states (b) mexico (c) countryside (d) atlas

**A:** The answer must be something that used to do what Google Maps and GPS services do, which is to give directions. Of the above choices, only atlases are used to give directions. So the answer is (d).

**Q:** Before getting a divorce, what did the wife feel who was doing all the work? Answer Choices: (a) harder (b) anguish (c) bitterness (d) tears (e) sadness

**A:** The answer should be the feeling of someone getting divorced who was doing all the work. Of the above choices, the closest feeling is bitterness. So the answer is (c).

---

Table 12: Few-shot exemplars for full chain of thought prompt for AQuA when backward verification.

---

**PROMPT FOR AQUA ALGEBRAIC WORD PROBLEMS**

**Q:** "John found that the average of 15 numbers is 'X'. If 10 is added to each number then the mean of the numbers is 50." What is the answer of 'X'?

**A:** If 10 is added to each number, then the mean of the numbers also increases by 10. The new mean would be 50., X + 10 = 50, X = 40. The answer is 40.

**Q:** "If a / b = 'X' and 8a + 5b = 22, then the value of a is 3/2." What is the answer of 'X'?

**A:** If a / b = X, then 8a + 5b = 22 and a = 3/2, so , 8 * 3/2 + 5b = 22, 5b = 22 - 12 = 10, b = 2, X = a / b = 3/2 / 2 = 3/4. The answer is 3/4.

**Q:** "A person is traveling at 'X' km/hr and reached his destiny in 2.5 hr then find the distance is 50km." What is the answer of 'X'?

**A:** The distance that the person traveled would have been , X km/hr * 2.5 hrs = 50 k, X = 20. The answer is 20.

**Q:** "There were 'X' computers in the server room. Five more computers were installed each day, from monday to thursday. There are 29 computers in the server room." What is the answer of 'X'?

**A:** There were originally X computers. For each of 4 days, 5 more computers were added. So 5 * 4 = 20 computers were added. And there are 29 computers. X + 20 = 29, X = 29 - 20, X = 9. The answer is 9.

---

Table 13: Few-shot exemplars for full chain of thought prompt for Date Understanding when backward verification.

---

**PROMPT FOR DATE UNDERSTANDING**

**Q:** "'X' is coming in 36 hours. One week from today is 01/05/2015." What is the answer of 'X'?

**A:** ==If The date one week from today is 01/05/2015, so today is 12/30/2014. So the data after 36 hours is 2015.== The answer is 2015.

**Q:** "The first day of 'X' is a Tuesday, and today is the first Monday of 2019. Today is 01/07/2019." What is the answer of 'X'?

**A:** ==If today is the first Monday of 2019 and today is 01/07/2019. So The first day of 2019 is a Tuesday.== The answer is 2019.

**Q:** "The concert was scheduled to be on 'X'/01/1943, but was delayed by one day to today. 10 days ago is 05/23/1943." What is the answer of 'X'?

**A:** ==10 days ago is 05/23/1943, and the concert was delayed by one day to today, so today is 06/02/1943. So the concert was scheduled to be on 06/01/1943== The answer is 06.

**Q:** "It is 'X'/19/1969 today. 24 hours later is 04/20/1969." What is the answer of 'X'?

**A:** ==24 hours later is 04/20/1969. So today is 04/19/1969.== The answer is 04.

**Q:** "Jane thought today is 'X'/12/2002, but today is in fact Mar 12, which is 1 day later. 24 hours later is 03/13/2002." What is the answer of 'X'?

**A:** ==24 hours later is 03/13/2002. So today is 03/12/2002.== The answer is 03.

**Q:** "Jane was born on the last day of Feburary in 'X'. Today is her 16-year-old birthday. Yesterday is 02/27/2017" What is the answer of 'X'?

**A:** ==Yesterday is 02/27/2017, so today is 02/28/2017, Jane was born on 02/28/2001.== The answer is 2001.

---

Table 14: Few-shot exemplars for full chain of thought prompt for math word problems when backward verification. This set of exemplars was used for all math word problem datasets except AQuA.

---

**PROMPT FOR MATH WORD PROBLEMS**

**Q:** "There are 'X' trees in the grove. Grove workers will plant trees in the grove today. After they are done, there will be 21 trees. The grove workers planted 6 trees today.What is the answer of 'X'?

**A:** There are X trees originally. The grove workers planted 6 trees today. Then there were 21 trees after some more were planted. So, we can write the following equation:X + 6 = 21, X = 21 - 6, X = 15. The answer is 15.

**Q:** "If there are 'X' cars in the parking lot and 2 more cars arrive, There are 5 cars in the parking lot." What is the answer of 'X'?

**A:** There are originally X cars. 2 more cars arrive and there are 5 cars finally. So: X + 2 = 5, X = 5 - 2, X = 3. The answer is 3.

**Q:** "Leah had 'X' chocolates and her sister had 42. If they ate 35, they have 39 pieces left in total." What is the answer of 'X'?

**A:** Originally, Leah had X chocolates. Her sister had 42. So in total they had: X + 42 = Y. After eating 35, they had 39, so, Y = 35 + 39, Y = 74, X + 42 = 74, X = 74 - 42, X = 32. The answer is 32.

**Q:** "Jason had 20 lollipops. He gave Denny some lollipops. Now Jason has 'X' lollipops. Jason gave Denny 8 lollipops." What is the answer of 'X'?

**A:** Jason started with 20 lollipops. Then he had X after giving some to Denny and gave Denny 8. 20 - X = 8, X = 12. The answer is 12.

**Q:** "Shawn has 'X' toys. For Christmas, he got two toys each from his mom and dad. He has 9 toys now." What is the answer of 'X'?

**A:** Shawn started with X toys. If he got 2 toys each from his mom and dad, then that is 4 more toys. X + 4 = 9, X = 9 - 4, X = 5. The answer is 5.

**Q:** "There were 'X' computers in the server room. Five more computers were installed each day, from monday to thursday. There are 29 computers in the server room." What is the answer of 'X'?

**A:** There were originally X computers. For each of 4 days, 5 more computers were added. So 5 * 4 = 20 computers were added. And there are 29 computers. X + 20 = 29, X = 29 - 20, X = 9. The answer is 9.

**Q:** "Michael had 58 golf balls. On tuesday, he lost 'X' golf balls. On wednesday, he lost 2 more. He had 33 golf balls at the end of Wednesday." What is the answer of 'X'?

**A:** Michael started with 58 golf balls. After losing X on tuesday and he lost 2 more on wednesday, He had 33 golf balls. So, we can write the following equation: 58 - X - 2 = 33, 58 - X = 35, X = 23. The answer is 23.

**Q:** "Olivia has $'X'. She bought five bagels for $3 each. She has 8 dollars left." What is the answer of 'X'?

**A:** Olivia had X dollars. 5 bagels for 3 dollars each will be 5 x 3 = 15 dollars. She has 8 dollars left finally. X - 15 = 8, X = 8 + 15, X = 23. The answer is 23.

---

Table 15: Few-shot exemplars for full chain of thought prompt for CSQA when backward verification. There are newlines between the answer choices that are omitted in the table for space reasons.

---

**PROMPT FOR CSQA**

**Q:** "People use blotter to absorb extra ink from a fountain pen." Judge whether this statement is normal (yes or no).

**A:** The Blotter is used to absorb extra ink from a fountain pen. The answer is Yes.

**Q:** "Television requires cable." Judge whether this statement is normal (yes or no).

**A:** The Television is an electrical appliance, it needs electricity, so it requires cable. The answer is Yes.

**Q:** "The fox walked from the city into the forest, it was looking for a hen house." Judge whether this statement is normal (yes or no).

**A:** The hen house is not in the forest, so the fox does not go to the hen house. The answer is No.

**Q:** "Sammy wanted to go to where the people were. He might go populated areas." Judge whether this statement is normal (yes or no).

**A:** There are many people in the populated areas, so they really go here. The answer is Yes.

**Q:** "The grapes are put in the fruit market just before checking out." Judge whether this statement is normal (yes or no).

**A:** The answer should be the place where grocery items are placed before checking out. But the fruit market is not suitable place where grocery items are placed. The answer is No.

**Q:** "Google Maps and other highway and street GPS services have replaced the united states." Judge whether this statement is normal (yes or no).

**A:** The united states is a country and Google Maps is a map, so Google Maps cannot replace the united states. The answer is No.

**Q:** "The wife who was doing all the work felt bitterness before getting a divorce." Judge whether this statement is normal (yes or no).

**A:** The wife divorced who was doing all the work. So she felt bitterness. The answer is Yes.

---

Table 16: Few-shot exemplars for full chain of thought prompt for AQuA when True-False Item Verification.

---

**PROMPT FOR AQUA ALGEBRAIC WORD PROBLEMS**

**Q:** "John found that the average of 15 numbers is 40. If 10 is added to each number then the mean of the numbers is 50." Do it is correct (True or False)?

**A:** The new mean would be 50. The average of 15 numbers is 4, if 10 is added to each number, then the mean of the numbers also increases by 10. 50 - 40 = 10. The answer is True.

**Q:** "If a / b = 3/4 and 8a + 5b = 22, then the value of a is 3." Do it is correct (True or False)?

**A:** If a is 3, a / b = 3/4, so b = 4. then 8a + 5b = 8 * 2 + 5 * 4 = 36, but 8a + 5b = 22 The answer is False.

**Q:** "A person is traveling at 20 km/hr and reached his destiny in 2.5 hr then find the distance is 65km." Do it is correct (True or False)?

**A:** If 65km is driven at 20km/hr, so the driving time is 65km / 20km/hr = 3.25h, but he destiny in 2.5 hr. The answer is False.

**Q:** "There were 9 computers in the server room. Five more computers were installed each day, from monday to thursday. There are 29 computers in the server room." Do it is correct (True or False)?

**A:** There are 29 computers in the server room. For each of 4 days, 5 more computers were added. 5 * 4 = 20 computers were added. So there were originally 9 computers. The answer is True.

---

Table 17: Few-shot exemplars for full chain of thought prompt for math word problems when True-False Item Verification. This set of exemplars was used for all math word problem datasets except AQuA.

---

**PROMPT FOR MATH WORD PROBLEMS**

**Q:** 'There are 15 trees in the grove. Grove workers will plant trees in the grove today. After they are done, there will be 21 trees. The grove workers planted 4 trees today.' Do it is correct (True or False)?

**A:** If the Grove workers will plant 4 trees today and there will be 21 trees after they are done. 21 - 4 = 17, there are 17 trees in the grove, but actually there are 15 trees, 17 != 15, which is different from the theme. The answer is False.

**Q:** 'If there are 3 cars in the parking lot and 2 more cars arrive, There are 5 cars in the parking lot.' Do it is correct (True or False)?

**A:** If there will be 5 cars in the parking lot, subtract 2 cars that will arrive, 5 - 2 = 3, so there are 2 cars in the parking lot, which is consistent with the theme. The answer is True.

**Q:** 'Leah had 32 chocolates and her sister had 42. If they ate 35, they have 39 pieces left in total.' Do it is correct (True or False)?

**A:** If there are 39 pieces of chocolates and 35 pieces of chocolate are eaten, Leah and her sister have 39 + 35 = 74 in total. Her sister's had 42, so Leah had 74 - 42 = 32, which is consistent with the theme. The answer is True.

**Q:** 'Jason had 20 lollipops. He gave Denny some lollipops. Now Jason has 12 lollipops. Jason gave Denny 6 lollipops.' Do it is correct (True or False)?

**A:** If Jason gave Denny 6 lollipops, and Jason now has 12 lollipops, so Jason originally had 6+12=18 lollipops, 18 != 20, which is different from the theme. The answer is False.

**Q:** 'Shawn has five toys. For Christmas, he got two toys each from his mom and dad. He has 9 toys now.' Do it is correct (True or False)?

**A:** If Shawn now has 9 toys and his parents gaven him two each, then he originally had 9 - 2 - 2 = 5, which is consistent with the theme. The answer is True.

**Q:** 'There were nine computers in the server room. Five more computers were installed each day, from monday to thursday. There are 18 computers in the server room.' Do it is correct (True or False)?

**A:** Now there are 18 computers in the server room. For each of 4 days, 5 more computers were added. So 5 * 4 = 20 computers were added. So there were 18 - 20= -2 in the server room originally, -2 != 9, which is different from the theme. The answer is False.

**Q:** 'Michael had 58 golf balls. On tuesday, he lost 23 golf balls. On wednesday, he lost 2 more. He had 40 golf balls at the end of Wednesday.' Do it is correct (True or False)?

**A:** If Michael had 40 golf balls on Wednesday, he had 40+2=42 on Tuesday because he lost 2 golf balls on Wednesday. Due to lost 23 balls on Tuesday, he should have 42+23=65 on Monday, but in fact Michael has 58 golf balls original, which is different from the theme. The answer is False.

**Q:** 'Olivia has $23. She bought five bagels for $3 each. She has 8 dollars left.' Do it is correct (True or False)?

**A:** If Olivia had $8 left and she bought five bagels for $3 each, so costs 5 * 3 = 15, so there was 8 + 15 = 23, which is consistent with the theme. The answer is True.