# OpenReview forum: "Large Language Models are Better Reasoners with Self-Verification"
_EMNLP/2023/Conference — EMNLP 2023 Findings_

### Official Review · Reviewer_hdXu · 2023-08-03

**Soundness:** 3

**Excitement:**

3: Ambivalent: It has merits (e.g., it reports state-of-the-art results, the idea is nice), but there are key weaknesses (e.g., it describes incremental work), and it can significantly benefit from another round of revision. However, I won't object to accepting it if my co-reviewers champion it.

**Paper Topic And Main Contributions:**

This paper studies how to improve the chain of thought (CoT) prompting to enhance the reasoning abilities of large language models (LLMs). The main contribution of this paper is the proposal and demonstration of the self-verification abilities of large language models. Specifically, the authors introduce a self-verification method that includes a forward reasoning generation and a backward verification. Forward Reasoning involves LLM reasoners generating candidate answers using CoT.  In Backward Verification, the original condition is masked, and its result is predicted using another CoT. Candidate conclusions are ranked based on a verification score that assesses the consistency between the predicted and original condition values. The paper also presents experimental results that demonstrate the effectiveness of the proposed method in solving arithmetic, commonsense, and logical reasoning tasks.

**Reasons To Accept:**

1) This paper improves the chain of thought (CoT) prompting by introducing backward verification.
2) The proposed method allows LLMs to verify whether the generated CoTs are correct or not, simulating human thinking and verifying processes in solving complex reasoning problems.
3) Experimental results demonstrate the effectiveness of the backward verification on arithmetic, commonsense, and logical reasoning tasks.

**Reasons To Reject:**

1) The improvement over CoT baselines, especially Self-Consistency Decoding CoTs, is not very significant. With Self-Consistency Decoding, the average improvement is usually around 0.5%, which may limit the real application of the proposed method considering the higher computing usage of backward verification.
2) The method does not work very effectively on general reasoning tasks compared with mathematic reasoning.
3) Lack of deep analysis on when back verification would work and when would not. From the current results, we can only conclude that the proposed method may work better in mathematical reasoning tasks. One possible reason is the masked conditions might be more effective than True-False Item Verification. Some deep qualitative analysis is needed here, which may also further help to know how to generalize similar verification approaches to other reasoning tasks beyond arithmetic reasoning.

**Reproducibility:**

4: Could mostly reproduce the results, but there may be some variation because of sample variance or minor variations in their interpretation of the protocol or method.

**Reviewer Confidence:**

4: Quite sure. I tried to check the important points carefully. It's unlikely, though conceivable, that I missed something that should affect my ratings.

---

> ### Author Rebuttal · Authors · 2023-08-26
>
> Thank you for your thorough and constructive review. We value your suggestions and have taken them into consideration. Here is our response to each point:
>
> ---
>
> > **Low Improvement Over the Baselines:**
>
> We apologize if we provided the impression that our proposed method possesses marginal gains over Self-Consistency Decoding. Because our goal is not to improve self-consistency, but to unlock the potential of LLMs compared to the basic baseline. The effect of our method is indeed more significant when used with simpler baselines (e.g., CoT) as opposed to sophisticated ones like self-consistency and PAL. That being said, the primary idea behind our paper was to emphasize the self-verification capability of LLMs, as a methodology to improve reasoning capability of the model. Regardless of baseline or the complexity of the method used for forward reasoning, we demonstrated that by introducing a simple backward verification process, the model's reasoning ability could still be further improved. As for the concern about the computational overhead, we showed that even when the number of backward verification processes (P) is set to a small value (e.g., P=2), a noticeable improvement can still be achieved.
>
> ---
>
> > **Does not work very effectively on general reasoning task:**
>
> We agree that, in our current work, the self-verification method perform better on mathematical reasoning than general reasoning tasks - this is indeed an area we will explore in future work. The primary difference lies in how the problems are set up: mathematical reasoning tasks are well-defined and the validation steps are relatively clear, while in commonsense reasoning tasks, there's more room for interpretation and ambiguity. our proposed method TFV (True-False Item Verification) does not perform as effectively as it does in arithmetic ones (CMV: Condition Mask Verification). We'll investigate approaches that could enhance the performance on general reasoning tasks in our future studies.
>
> ---
>
> > **Lack of Deeper Analysis:**
>
> We appreciate your suggestion that our paper could benefit from a deeper analysis of the conditions that make our proposed method effective and what might limit its applicability. We performed additional analysis in response to your suggestions.
>
> - 1. You are correct in noting that our proposed method seems to perform particularly well in mathematical reasoning tasks. The achievement is primarily due to the predictable structure of these tasks, allowing us to develop effective verification questions via condition masking. Mathematical reasoning tasks often involve well-defined variables and operators, and the relationships between variables are often explicit, making it relatively straightforward to mask conditions and generate verification questions.
>
> - 2. In our paper, we also apply the proposed method TFV to commonsense and logical reasoning tasks, which yielded relatively modest improvements over the baseline. Unlike arithmetic reasoning tasks, commonsense and logical reasoning tasks involve a greater variety of semantic structures and more complex reasoning procedures, which makes condition masking challenging.
>
> ---
>
> **We hope this extra analysis addresses your concerns. We truly appreciate your fruitful suggestion, and we will incorporate them in future versions of our paper.**

---

### Official Review · Reviewer_iNBL · 2023-08-03

**Soundness:** 4

**Excitement:**

4: Strong: This paper deepens the understanding of some phenomenon or lowers the barriers to an existing research direction.

**Paper Topic And Main Contributions:**

This work proposes to let LLMs self-verify through three steps: mask the values provided in the question; treat the answers generated in the forward inference as known conditions; ask the LLM to answer the backward question, that is, provide the masked value.

**Reasons To Accept:**

The self-verification method is novel, and verification results can improve performance.

**Reasons To Reject:**

Lack of analysis of self-verification accuracy, e.g. LLMs can return wrong answers to forward questions and also to generated backward questions;
The proposed method proceeds in an iterative manner resulting in high cost of querying the LLM.

**Reproducibility:**

4: Could mostly reproduce the results, but there may be some variation because of sample variance or minor variations in their interpretation of the protocol or method.

**Reviewer Confidence:**

4: Quite sure. I tried to check the important points carefully. It's unlikely, though conceivable, that I missed something that should affect my ratings.

---

> ### Author Rebuttal · Authors · 2023-08-26
>
> We appreciate the reviewer's insightful comments and the time devoted to reviewing our work. We understand the concerns raised and would like to address them as follows:
>
> ---
>
> > **Concern:** Lack of analysis of self-verification accuracy, e.g. LLMs can return wrong answers to forward questions and also to generated backward questions.
>
> **Response:** We agree this is an important aspect and have discussed it in Appendix A.4 and Table 6 in the manuscript. Our hypothesis is that even if an LLM might generate a wrong answer in the forward reasoning stage, we can still obtain more reliable results by evaluating its reasoning result via a backward verification phase. The results in Table 6 demonstrate that LLMs are more capable of selecting the correct answer from a set of incorrect candidates,  rather than selecting a wrong answer. We will add more discussion about this in the revised version.
>
> **Table 6**
>
> | Dataset | GSM8K | SingleEq | AddSub | |  MultiArith | AQUA-RAT | SVAMP |
> |---------|-------|----------|--------|-|------------|----------|-------|
> | Correct Conclusion | **77.3** | **96.7** | **91.4** | |  **97.3** | 64.0 | **84.3** |
> | Incorrect Conclusion | 72.4 | 64.4 | 51.9 | | 81.1 | **64.7** | 66.9 |
>
> ---
>
> > **Concern:** The proposed method proceeds in an iterative manner resulting in high cost of querying the LLM.
>
> **Response:** True, our method invokes the LLM more than once for each task, which would lead to higher computational costs compared to the one-step prediction approach. However, the increased costs are justified by the significant improvement in the performance across different tasks. In a practical scenario, when the accurate reasoning is prioritized, the additional computational overhead can be considered an acceptable trade-off. Moreover, our computational resource analysis (Figure 7 in the manuscript) has shown that even with a limited number of backward verification (e.g., P=2), our method still achieves superior performance to existing CoT baselines. Thus, users can flexibly choose the number of iterative steps according to their computational budget and the task requirement.
>
> ---
>
> **We value the inputs provided by the reviewer, and we are committed to refining and expanding the analysis to address the raised concerns.**

---

### Official Review · Reviewer_yvPK · 2023-08-05

**Typos Grammar Style And Presentation Improvements:** Line 444
**Soundness:** 3

**Excitement:**

3: Ambivalent: It has merits (e.g., it reports state-of-the-art results, the idea is nice), but there are key weaknesses (e.g., it describes incremental work), and it can significantly benefit from another round of revision. However, I won't object to accepting it if my co-reviewers champion it.

**Paper Topic And Main Contributions:**

This paper analyzes the problems that current LLMs is prone to in multi-step reasoning, that is, it is sensitive to a certain error and easy to cause error accumulation. The authors propose a method of backward verification after forward reasoning, which alleviates this phenomenon and verifies that LLMs itself has the ability to self-verify the generated results.

The main contributions of this paper are: (1) The ability of LLMs to self-verify results with the help of specific methods is verified; (2) The feasibility of the proposed method is verified on multiple LLMs and multiple mathematical, commonsense, and logical reasoning datasets. (3) Two methods of self-validation are proposed. True-False Item Verification for General Tasks in the backward verification stage and Condition Mask Verification based on the characteristics of Arithmetic Tasks.

**Questions For The Authors:**

Q1: The proposed method includes forward reasoning and backward verification, wherein K candidate answers are generated in the forward reasoning process, and P times of reverse verification will be performed for each answer. Is there any deduplication process in the process of obtaining K candidate answers?

Q2: If the answer to Q1 is no, can it be understood that this process of forward reasoning is equivalent to a self-consistency? (Equivalent to the effect shown by this method to a large extent with the strength of the idea of self-consistency, so this also explains why the effect is weak when SV and SC methods are combined)

**Reasons To Accept:**

1. The authors analyze the problem with the current method of inference using the prompting method, that is, LLMs may be sensitive to a certain error in multi-step inference and accumulate errors in subsequent inference.

2. The authors propose a method to solve this problem by using the self-verification ability of the model to enhance the reasoning ability of the model, which includes two stages: forward reasoning and backward verification.

3. The authors clearly show the process of their proposed method (Fig. 2).

4. The author conduct a detailed analytical experiment (Sec.5).

5. The method proposed by the author can be combined with some existing methods (CoT, self-consistency (SC) and PAL are demonstrated in the paper).

**Reasons To Reject:**

1. The benefits from backward process are not analyzed separately (after all, the forward reasoning process is like getting benefits from self-consistency).

2. The models used in the experiment are slightly outdated, and the experiment lacks analysis of new models, such as text-davinci-003 and gpt-3.5-turbo, and even gpt-4

**Reproducibility:**

4: Could mostly reproduce the results, but there may be some variation because of sample variance or minor variations in their interpretation of the protocol or method.

**Reviewer Confidence:**

4: Quite sure. I tried to check the important points carefully. It's unlikely, though conceivable, that I missed something that should affect my ratings.

---

> ### Author Rebuttal · Authors · 2023-08-25
>
> Thank you for your thoughtful feedback, we truly appreciate the recognition of our work and the insightful comments made. In the experiment, we tried our best to eliminate the positive effect of Self-Consistency. Therefore, in Table 1, we tried to use Self-Consistency as a forward inference and added an extra self-verification process in the backward process to show that independent self-verification alone was enough to bring additional performance gains. To address your questions and concerns:
>
> ---
>
> > *Q1:* "The proposed method includes forward reasoning and backward verification, wherein K candidate answers are generated in the forward reasoning process, and P times of reverse verification will be performed for each answer. Is there any deduplication process in the process of obtaining K candidate answers?"
>
> *A1:*  **Yes.** In the forward reasoning stage, we generate K candidate answers using sampling decoding. It's possible that the sampling process might result in same answers. Therefore, in the current experiment setting, direct de-duplication is performed.
>
> In the code section of the supplementary material (`code/main_verifier.py line 156`), we also demonstrate the process of eliminating duplicate data:
>
> ```
> if p_item not in answers: # For the duplicate candidate answers generated in the forward, we excluded them.
>     answers.append(p_item)
> ```
>
>
> For example:
> ```
> candidate_answers = forward_reason()
> print(candidate_answers)
> >>> [2,3,3,3,5]
>
> answers = []
> for p_item in candidate_answers:
>     if p_item not in answers: # For the duplicate candidate answers generated in the forward, we excluded them.
>         answers.append(p_item)
>
> print(answers)
> >>> [2,3,5]
>
> backward_reason(answers)
> ```
> ---
>
> > *Q2:* "If the answer to Q1 is no, can it be understood that this process of forward reasoning is equivalent to a self-consistency?"
>
> *A2:* In our method, the LLM performs backward verification to evaluate the correctness of each answer, returning a score that signifies the model's confidence in that specific answer. Our method further provides a verification mechanism for each predicted answer and ranks them based on the verification score, which cannot be done in the self-consistency method.
>
> ---
>
> We acknowledge the value of conducting experiments with newer models. In fact, due to the high cost of these models, we have primarily proceeded with code-davinci-001 and code-davinci-002. However, we agree that an exploration of the new models could offer additional insights into the scaling behavior of the self-verification method, and we plan to explore this in future work.
>
> Finally, we appreciate your attentive eye in catching the typo (PoL instead of PAL), and will make sure to correct this in the final version of the paper.
>
> **Your feedback is much appreciated and will certainly contribute to enhancing the quality of our work.**

---

### Meta-Review · Area_Chair_yC8B · 2023-09-15

**Recommendation:** 3

**Metareview:**

This paper proposes a method to improve reasoning by combining chain-of-thought with self-verification, e.g., its generated rationale is paraphrased with masks and the language model fills in it, which can be verified against the original value. This method is shown to consistently improve 6 arithmetic tasks and 2 general tasks.

Reviewers agreed that the paper's contribution is in a new method for self-verification (yvPK, iNBL, hdXu), the proposed method can be used together with existing methods like self-consistency and PAL which leads to the best performance (yvPK), and the paper includes comprehensive experiments with many datasets, baselines, and backbone LMs (yvPK, hdXu).

However, reviewers pointed out that there is insignificant gains, especially with self-consistency decoding and PAL, and on tasks beyond arithmetic tasks (hdXu). Also, they mentioned lack of analysis of the method (yvPK, iNBL, hdXu) and significant increase in cost (iNBL, hdXu) are weaknesses of the work. Authors are encouraged to address these issues in the final version of the paper if accepted.

---

### Decision · Program_Chairs · 2023-10-07

**Decision:**

Accept-Findings

**Comment:**

This paper proposes a method to improve reasoning by combining chain-of-thought with self-verification, e.g., its generated rationale is paraphrased with masks and the language model fills in it, which can be verified against the original value. This method is shown to consistently improve 6 arithmetic tasks and 2 general tasks.

Reviewers agreed that the paper's contribution is in a new method for self-verification (yvPK, iNBL, hdXu), the proposed method can be used together with existing methods like self-consistency and PAL which leads to the best performance (yvPK), and the paper includes comprehensive experiments with many datasets, baselines, and backbone LMs (yvPK, hdXu).

However, reviewers pointed out that there is insignificant gains, especially with self-consistency decoding and PAL, and on tasks beyond arithmetic tasks (hdXu). Also, they mentioned lack of analysis of the method (yvPK, iNBL, hdXu) and significant increase in cost (iNBL, hdXu) are weaknesses of the work. Authors are encouraged to address these issues in the final version of the paper if accepted.